# Is Foreign Direct Investment Resilient Post the COVID-19 Pandemic? The Case of a Subnational Economy

**Roxana Wright and Chen Wu \***

School of Business, Plymouth State University, Plymouth, NH 03264, USA; rwright01@plymouth.edu
\*   Correspondence: cwu@plymouth.edu; Tel.: +1-603-535-3241

**Abstract:** The disruption brought about by the COVID-19 pandemic has been unprecedented in its global reach and unique impacts. While the literature has addressed the disruption effect on FDI at the country level, we provide a unique dive into the presence and development of FDI at a subnational location. We use detailed data on spatial and industrial distributions of FDI in the U.S. state of New Hampshire and find support for all our hypotheses related to post-disruption recovery and resilience. Given the varied impact of the pandemic on FDI across locations, and the heterogeneity in local conditions, we contend that the subnational recovery depends on the impact of the disruption and happens at varying levels and timelines. While the literature documented that foreign businesses choose to embed in their local host environments, few studies have considered empirically how the level of local integration affects FDI recovery after disruption. We propose that subnational locations with a high level of integration maintain relative strength in FDI post-disruption. The COVID-19 pandemic disruption presents an opportunity to evaluate FDI resilience. We postulate that existing FDI and spatial agglomerations of FDI-related activities impact the post-disruption resilience of FDI at a subnational location. The analysis concludes on actionable insights for researchers and practitioners regarding how to navigate the FDI inflows and activities at their specific location.

**Keywords:** COVID-19; post-disruption recovery; FDI; subnational economy; resilience

## 1. Introduction

In 2020, the COVID-19 pandemic caused an unprecedented collapse in global foreign direct investment (FDI), and for the first time, the U.S. lost its position as the world's largest FDI recipient (in terms of the annual FDI inflows) to China. The former recorded a near 50 percent drop to USD 134 billion in FDI inflows, while the latter recorded a 4 percent increase to USD 163 billion (UNCTAD 2021). A strong boom in the global FDI kicked off in 2021 when many countries recorded FDI inflows that surpassed their pre-pandemic level. Although China continued to grow in its FDI inflows by 20 percent reaching a record USD 179 billion, the U.S. returned to the top of the list of recipients with USD 323 billion, an increase of 114% from the last year (UNCTAD 2022).

In the first three quarters of 2022, the FDI inflows in the U.S. totaled USD 233.4 billion, which remained the world's largest recipient of FDI. Figure 1 below displays the quarterly data on the U.S. FDI inflows. In the year before the COVID-19 pandemic, the U.S. attracted FDI at an average amount of USD 70 billion every quarter. This number declined to below USD 20 (USD 40) billion in the first (second) half of 2020 due to the lock-down of economies caused by the pandemic. It bounced up to more than USD 70 billion in the first half of 2021 and then rose sharply to more than USD 120 billion in the second half. Entering 2022, the U.S. quarterly FDI inflows returned to slightly above its pre-pandemic normal and remained in the values from USD 74 billion to USD 84.8 billion.

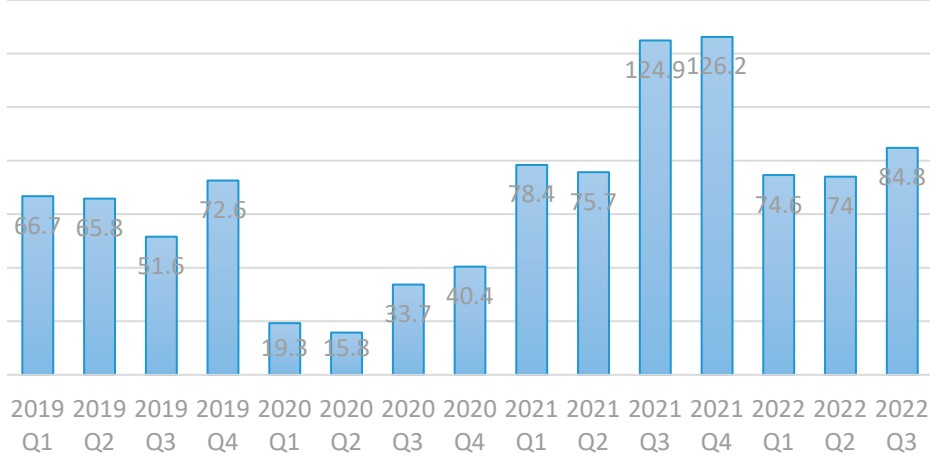

**Figure 1.** FDI Inflows (in Billion USD) in the U.S. by Quarter, 2019–2022 (Data Source: U.S. Bureau of Economic Analysis, U.S. International Transactions Data).

On the one hand, the big picture of global investment reveals its volatile nature across the countries, as discussed above. On the other hand, there exists significant heterogeneity in both the effects of the global pandemic disruptions and the outcomes of foreign businesses across the locations within a country (Wright and Wu 2022). For example, as a result of the COVID-19 pandemic, 43 states suffered declines ranging from −20.7 percent (Alaska) to −0.2 percent (Arizona) in the employment by majority-owned foreign firms in 2020, while 7 states either maintained (Arkansas) or witnessed increases ranging from 0.3 percent in Maine to 8.4 percent in Delaware (BEA 2023b). By the tail-end of the pandemic in 2021, the majority-owned foreign firms accounted for 6.2 percent of total private-sector employment of the US, but for New Hampshire, that share was 8.7 percent, which ranks the Granite State Number 3 among all 50 states (BEA 2023a). Therefore, a more local lens of investigation would shed light on understanding the status and outcomes of FDI in a locality, and thus provides meaningful implications for the local researchers and policy makers.

By presenting the local specifics of FDI activities in New Hampshire, US, the purpose of this study is to highlight the importance of FDI in local development and to determine how resilient the FDI at a subnational locality could be after the pandemic disruptions. The results of our investigation reveal how the COVID-19 pandemic impacted FDI at the subnational location. Comparisons are drawn with national patterns. The resulting hypotheses and the data supporting them show important economic aspects after the pandemic that are essential for decision making, both for strategists and researchers. The results also have valuable implications for both policy makers and researchers who are interested in attracting and promoting FDI to boost the local economies.

## 2. Literature Review

The theoretical foundation of our study merges two lines of research. The first is the conceptual models of FDI. Markusen (1984) modeled the Horizontal FDI as market-seeking investors who are driven by access to greater market potential and thus choose to invest in larger economies. The empirical studies that support the Horizontal FDI model report a positive relationship between market potential and FDI activities (Kumari and Sharma 2017; Gross and Ryan 2008; List et al. 2004). In contrast, Helpman and Krugman (1985) modeled the Vertical FDI as foreign investors who produce goods in the host market and then re-import the products back to the country of origin or export them to other countries. This model is supported by empirical studies that report a negative effect of the host market size (Braconier et al. 2005; Fredriksson et al. 2003; Rogers and Wu 2012). The second line of theoretical research is the FDI location choice theory based on Tinbergen's (1962) gravity model. Studies that adopted this model or its variations predict that the larger the sum

of host and origin countries' economic size, the larger the associated trade flows or FDI activities (Tuan and Ng 2007).

The relationship between location-specific characteristics and FDI has been the subject of recent research, with many conclusions on how local conditions and economic activity impact inward FDI (Nguyen et al. 2022; Monaghan 2014; Rogers and Wu 2012). As this view gains traction, researchers argue that existing frameworks for FDI analysis are still applicable, and that the only thing that changes is the perspective: economic, institutional, and political factors affect FDI levels and motivations at the subnational levels such as states (Batschauer da Cruz et al. 2022; Wu and Burge 2018), metropolitans (Castellani et al. 2022), and even cities (Danes et al. 2023; McDonald et al. 2018).

The recent literature attempts to investigate determinants and distributions of inward FDI across subnational locations. For instance, a panel-data study across states in India explored the FDI inflows and related them to subnational variables such as corruption, legal environment, business relationships, labor availability, political risk, intellectual property protection and agglomeration. Goswami (2023) found that FDI inflows are in fact determined by subnational institutional environments. Agglomeration also affects how FDI is distributed across states in India. The diversity of how and why FDI flows into subnational locations has been explored from more specific perspectives, with a focus on particular institutions, economic or social situations. For example, Karst and Johnson (2022) have found that political environment characteristics (such as majority political affiliation) across subnational location affects the distribution and type of FDI.

Some researchers are predicting that the COVID-19 pandemic intensified the uncertainty of internationally fragmented production processes, and, to some extent, proved the risks associated with highly dispersed global value chains. Extant studies (such as Zavarská 2022) anticipate that the FDI flows are likely becoming more regionalized and go towards locations that give specific strengths to value chains and add to investing companies' competitiveness, such as those with investment promotion legislation and meaningful infrastructure improvements. Research suggests that the determination of the FDI footprint post COVID-19 pandemic may be more nuanced (Wright 2022a), depending on the scope of the FDI-related activity and type of value chain.

Much of the literature showed that multinational companies tend to use local networking, prefer geographically concentrated investments, and engage in a variety of partnering with local actors, including subnational governments (Rosa et al. 2020). This type of local integration is common and appears to have a positive impact in how foreign subsidiaries generate value and profits (Lundan and Cantwell 2020). We can then assert that multinational companies' subsidiaries and their FDI activities will remain strong during and post disruption if there is robust local integration.

The literature showed that FDI is determined by location-specific factors. Since inflow, pattern, and distribution of FDI vary not only across countries and regions but also across subnational localities, it can be expected that local characteristics may also impact the strength of FDI activity post-disruption. In a study on how FDI inflows changed during disease outbreaks and epidemics, Yu et al. (2023) confirmed the shock to the economy and industries and found evidence of "compensatory FDI" after the disruption. The extent of the impact on FDI and level of compensatory FDI appeared to depend on local conditions and institutions. While their study focused on medical-related institutions, it lends legitimacy to the expectation that post-shock FDI recovery will vary across locations and be dependent on local situations.

Resilience is at the forefront of the thinking of decision makers and business professionals. The literature has long been concerned with the foreign subsidiary's performance and how the multinational companies take different gains from operating at varied international locations. The host-country effects on performance are documented, with more recent research proving the subnational home location of region's effects on performance of the subsidiary (Ma et al. 2013; Wu and Rogers 2018). These works indicated that the subnational location's characteristics shape the investing company's strategy and performance.

Drawing from such literature, we can expect that what FDI-generating companies have had and done before a significant disruption impacts their approaches during the shock and their post-disruption resilience.

The prevalence of companies to choose investments' locations that are geographically close to other related companies or beneficial partners has been documented in the literature. This co-location is associated with knowledge spillovers and a lowering of the liability of foreignness, leading to better performance outcomes (Rogers and Wu 2012). This phenomenon has been observed to be particularly prominent in situations with higher-than-usual uncertainty, such as investments in emerging economies after market liberalization (Lamin and Livanis 2013). It can then be alleged that, during disruption, the agglomeration effects will also be a development that FDI-producing companies will seek.

### 3. Methodology

Our study aims to interpret some FDI characteristics and resilience at a specific subnational location. Given the recency of the COVID-19 disruption, we focus on providing evidence from one in-focus subnational location and compare the circumstances and outcomes with national level data. Our scope is not to generalize determinant factors of resilience, but rather to observe and propose themes. To do this, we employ descriptive statistics and comparative analysis.

The present in-depth local level investigation focuses on the state of New Hampshire data to investigate local FDI levels and recovery patterns at a subnational location, and to place the findings against national data. The research first analyzes the aggregate data on investment expenditures and employment by the new FDI in New Hampshire in 2021. It then examines the FDI in New Hampshire from the perspective of presence of foreign subsidiaries. The analyzed sample contains 361 firms that are subsidiaries of parent companies outside the United States identified in several databases including the Uniworld Online database and the fDi Markets database, verified by the authors via visiting the company websites and phone contacts. The data for other key variables are collected from various publicly available sources with citations provided throughout the study. The distributions of foreign subsidiaries are analyzed across industries, value chain activities, and the employment sizes. The findings provide valuable inferences on contributions that foreign firms make to the state.

Our research also scrutinizes the location choices of parent companies with focuses on what countries they are from, in what industries they operate, what counties in New Hampshire they have subsidiaries, and how their subsidiaries fit in the local supply chain. The last part of our analysis is an interesting comparison of the foreign business status in New Hampshire between 2018 and 2022. The comparison adds to the efforts to understand the short-run effects of the COVID-19 pandemic on FDI at the subnational level.

The comprehensive descriptive analysis of both the new FDI and the presence of foreign subsidiaries at the level of counties, sectors and industries of New Hampshire provides unique evidence on the integration of foreign businesses in the ecosystem of New Hampshire. It also contributes to the study of FDI location choices by mapping the countries of origin for FDI in the various aspects of New Hampshire economy. The results hold important value for identifying the sources of FDI, the geographic clusters and industrial agglomerations of foreign businesses in New Hampshire, the sectors and industries where foreign firms add jobs and the extent to which foreign firms strengthen the state's local supply chain. In addition, because our study tracks the same sample of foreign subsidiaries from the pre-pandemic year of 2018 to the tail end of the pandemic in 2022, the findings provide consistent evidence on the contribution of FDI to the New Hampshire economy and the resilience of foreign businesses in New Hampshire despite facing challenges from the global disruptions.

### 4. Hypotheses

The present study intends to understand aspects related to FDI at a subnational location in the context of disruption. We are particularly interested in interpreting whether the local situation can play a role in how FDI is restored at pre-disruption levels, how foreign businesses' integration gives them strength to withstand disruption, and if there is a pattern between foreign business agglomeration and FDI resilience. Our aim is to propose several themes of FDI disruption and resilience.

As the literature showed, FDI determinants are different across national locations in the post-COVID-19 era (Al-Kasasbeh et al. 2022). Since the determinants differ, it can be expected that the effects of disruption on FDI inflows also vary. Recent analysis supports a pattern of unequal impact of global disruption across subnational locations (Wright and Wu 2022). Post-disruption, the recovery will be determined by all these patterns: existing FDI inflows pre-pandemic, FDI fluctuations during the shock, and speed and pattern of FDI recovery. Cheung et al. (2022) evaluated the extent of pandemic arbitrage resulting in reallocation of FDI flows and found that "FDI flows declined to destination markets that performed worse than source markets in COVID-19 infection rates, with the effect more evident in greenfield FDI".

Since the impact of the COVID-19 pandemic has been different across locations, and conditions leading to FDI inflows are heterogenous across locations, we propose that the recovery of FDI also vary. In the subnational context, one can therefore expect that the recovery of incoming FDI flows vary within the national boundaries. So, we content that:

**Hypothesis 1.** *The global disruption will have more severe impacts on new incoming FDI for some subnational locations than others. So, the new FDI activities in some locations will take a longer-than-national average time to restore their pre-disruption levels.*

As more companies engage in international and global FDI activities, it is imperative for the literature to recognize how these companies react to disruption and how their actions impact overall FDI inflows at a location. In this vein of research, it has been suggested that many multinational companies may suffer from an overstretch of their geographical footprint. These organizations may expand their reach beyond what is sustainable given their capabilities. Research on the topic before the pandemic found that regional integration of FDI locations across the value chain can alleviate the effects of this overstretching on the company's performance (Tang and Ruigrok 2022).

The idea of integration and embeddedness in the local environment is not new, but recent studies provided insights into how multinational corporations act when bringing FDI into a location. Song (2022) found that, under conditions of demand uncertainty, FDI activities tend to be less locally embedded, so as to give the company more flexibility. However, this finding and the level of local integration may be location dependent. Antalóczy et al. (2022) described that locations with lower levels of FDI embeddedness suffer in terms of what they can offer for development of local and incoming businesses. The authors suggested that "increasing the embeddedness of FDI already present" is key to fostering stability and attracting new projects.

Local integration of multinational corporations is expected to benefit companies engaging in FDI-related activities. Bussy and Zheng (2023) discovered that the responses of FDI to geopolitical risks are nuanced by information gathered through embeddedness. Companies with closer geographic, cultural, and commercial ties appear to be better informed and navigate these types of risks differently from companies with looser ties. In the post-disruption context, we contend that FDI levels are likely to be maintained at subnational locations where FDI-generating companies are more integrated in the local value chain structures or are more closely related to other businesses and institutions. Hence, we posit the following:

**Hypothesis 2.** *In the post-disruption era, FDI-related activities are likely to remain strong at a subnational location where foreign businesses are integrated with domestic businesses, either in value chains or business ecosystems.*

Many studies show the connection between FDI volumes and inflows on the one hand and the FDI geographical concentration. Ramachandran and Sasidharan (2022) recently looked at this connection in a country with highly concentrated FDI geographical distribution. Their empirical analysis found that "prior investment has a key role in the location choice of new investments. Further, the country of origin and industry sector of prior investments is associated with the location choice of new investors", resulting to an agglomeration of FDI at particular locations.

Industry clusters and businesses' spatial agglomeration have long been recognized to create benefits for the performance of companies and the economic development of their host locations (Rosenfeld 1997). The relationship between multinational companies' FDI-related activities, the local environment characteristics and business resilience has been documented recently (Wright 2022b). The presence of other foreign firms and existing FDI may be a significant determinant of FDI resilience.

We believe that the recent COVID-19 pandemic disruption offers a unique opportunity to evaluate FDI resilience during and post shock and propose that presence of FDI originating companies and geographic agglomerations of FDI-related activities determine the resilience of FDI at a subnational location (Arregle et al. 2009). Thus, we put forward:

**Hypothesis 3.** *The FDI presence tends to be resilient at a subnational location that has established strong ties with the FDI origins and where there is a relatively high level of foreign business agglomeration from the pre-disruption era.*

## 5. Data Analysis and Findings

To investigate themes of FDI recovery and resilience put forth in our hypotheses, we interpreted FDI presence and activity related data at a subnational location. In focus for our study was the U.S. state of New Hampshire. Since our goal was not to generalize on determining factors of FDI recovery or resilience, but rather to interpret these aspects at a subnational location, we examine the patterns of data and identify themes related to our three hypotheses, specifically on recovery of new FDI, local integration and FDI resilience, and FDI presence and resiliency at the subnational level, respectively.

### 5.1. Recovery of New FDI at the Subnational Level

For the first hypothesis, we examine the new FDI data for the state of New Hampshire and compare it with the U.S. new FDI patterns. The newly acquired, established, or expanded U.S. businesses by foreign enterprises (new FDI, hereafter) in New Hampshire are recovering from the COVID-19 Pandemic. As shown in Figure 2, the new FDI located in New Hampshire made a total investment expenditure of USD 5 million in 2021, equaling 2.5 times of the 2020 value during the COVID-19 pandemic when only the newly established US affiliates made an investment expenditure of USD 2 million. However, as shown in Figure 3 below, the first-year investment expenditures of new FDI in New Hampshire used to be averaged at USD 148 million between 2016 and 2019. This comparison highlights a cliff-like drop in the state's new FDI brought about by the COVID-19 pandemic, and that the new FDI in the state is still far below its pre-pandemic level as of 2021.

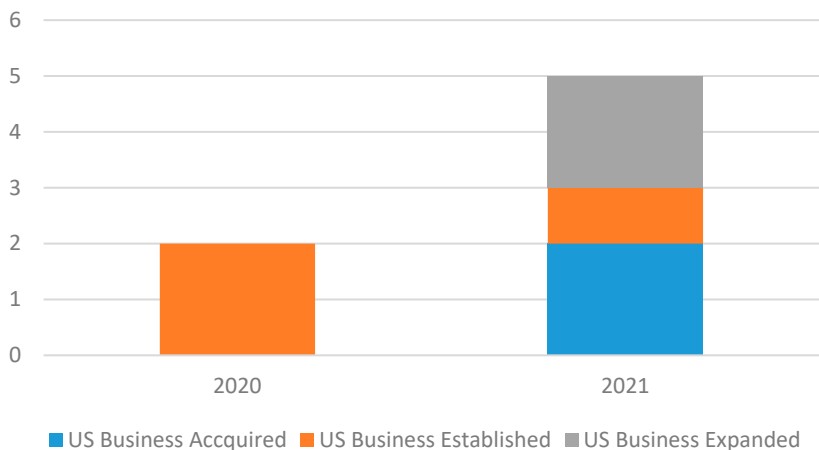

**Figure 2.** First-year Investment Expenditures (million USD) of New FDI in New Hampshire, 2020–2021 (Data Source: U.S. Bureau of Economic Analysis).

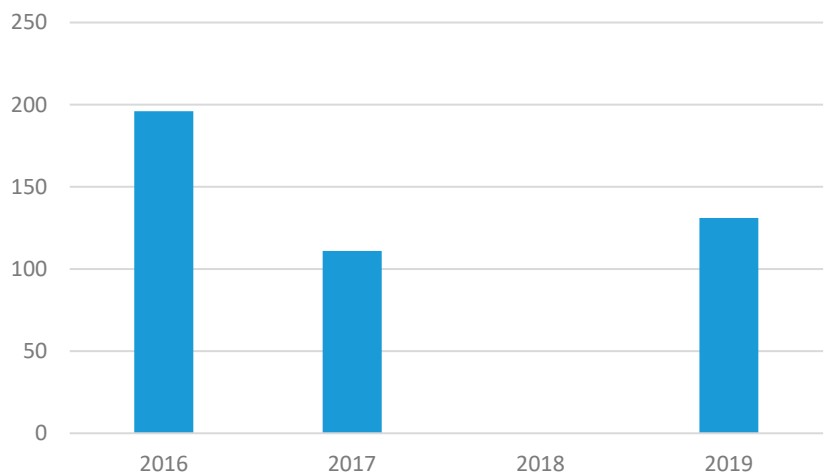

**Figure 3.** First-year Investment Expenditures (million USD) of New FDI in New Hampshire, 2016–2019 (Data Source: U.S. Bureau of Economic Analysis. The 2018 data were suppressed to avoid disclosure of data of individual companies).

Because the gross investment spending, by the domestic investors or by the foreign investors, tends to be volatile over time, it is worth comparing the New Hampshire data with country measures. At the national level, which is shown in Figure 4, although the COVID-19 pandemic in 2020 also caused the lowest first-year investment expenditures of the country's new FDI since 2014, dropping from USD 440 billion in 2015 to only USD 141 billion in 2020, this number drastically bounced up by 2.4 times to USD 334 billion. It is worth noting that the U.S. and New Hampshire shared the same percentage growth in the new FDI expenditures from 2020 to 2021 but the U.S. restored and outperformed the average of its values between 2014 and 2019, while New Hampshire did not. This comparison reveals the COVID-19 pandemic had a more severe impact on some states (including New Hampshire) than the others. Figure 4 also reveals the declining trend in the investment expenditures of the Greenfield new FDI (or the newly established U.S. affiliates) in the US, which has been going down every year from USD 14 billion in 2014 to USD 1.6 billion 2021.

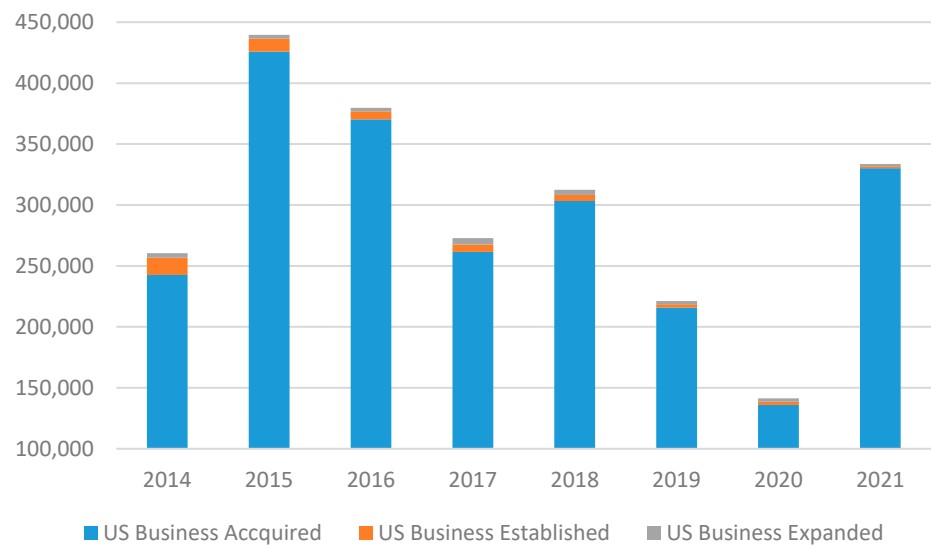

**Figure 4.** First-year Investment Expenditures (million USD) of New FDI in the US, 2014–2021 (Data Source: U.S. Bureau of Economic Analysis).

New Hampshire has also been recovering from the 2020 COVID-19 pandemic in the sense of job creations by the new FDI. As shown in Table 1 below, 100 jobs were planned to be created by the newly expanded US affiliates in 2021, compared to zero job creation by the new FDI one year ago. In the years before the pandemic, job creations by the state's new FDI were volatile and fluctuating between 100 and 1000 jobs every year, mainly created by the newly acquired US affiliates. But the relevant importance of the newly expanded US affiliates in supporting the New Hampshire employment has been rising since 2019. The newly established US affiliates have been negligible contributors to the employment of the state.

**Table 1.** Planned Employment by the New FDI in New Hampshire, 2014–2021 (1000 Employees).

|  | 2014 | 2015 | 2016 | 2017 | 2018 | 2019 | 2020 | 2021 |
|---|---|---|---|---|---|---|---|---|
| Total Planned employment | 0.9 | 0.1 | 1 | 0.4 | 0.1 | F | (*) | 0.1 |
| US Business Acquired | 0.8 | 0.1 | 1 | 0.3 | 0.1 | A | 0 | 0 |
| US Business Established | 0 | 0 | 0 | (*) | (*) | (*) | (*) | (*) |
| US Business Expanded | 0.1 | 0 | 0 | 0 | 0 | F | 0 | 0.1 |

Data source: U.S. Bureau of Economic Analysis. The size ranges are: A–1 to 499; F–500 to 999. (*) means a nonzero value that rounds to zero.

The national situation looks similar to New Hampshire. Figure 5 below illustrates that job creations by the new FDI at the national level was declining annually since 2017 (512,000 employees) to 2020 (222,000 employees). It slightly increased to 234,500 jobs in 2021 but was still lower than the year of 2019, whose value (299,000 employees) was the lowest in the pre-pandemic era. A closer scrutiny of the data reveals that while the planned employment by the newly acquired U.S. affiliates in 2021 was higher than the pandemic year, the employment of both the newly established and the newly expanded U.S. affiliates continued to worsen (from 11,600 to 3800, and from 7800 to 3100, respectively). This is opposite to New Hampshire, where newly expanded U.S. affiliates became increasingly important for the state's employment expansion.

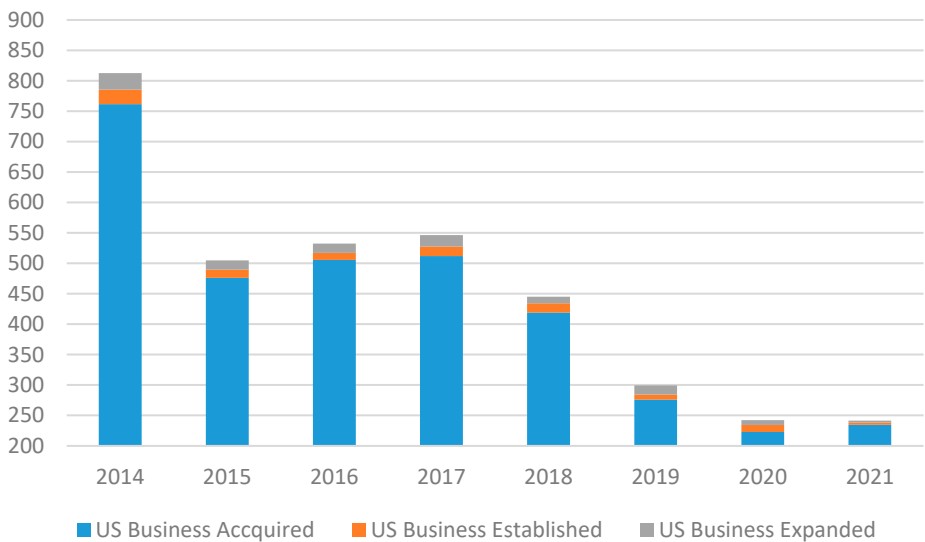

**Figure 5.** Planned Employment (1000 employees) by the New FDI in the U.S., 2014–2021 (Data Source: U.S. Bureau of Economic Analysis).

The above examination shows that, based on the current data, the nuances of levels and types of the new FDI, as well as the evolution of new FDI activities at a subnational location juxtaposed to the national-level data. These data support our first hypothesis, indicating that global disruption will have varying impacts on new incoming FDI across subnational locations. New FDI activities at a subnational location can thus be anticipated to take a longer-than-national average time to restore their pre-disruption levels.

*5.2. Local Integration and FDI Resilience*

Analysis supporting the second hypothesis is based on data on characteristics of FDI-generating companies and their activities. The data and the examination are focused on the state of New Hampshire, as a relatively lesser destination of FDI inflows volume nationally. Nevertheless, even in this circumstance, the findings support the importance of subnational FDI activities integration in local value chains and business environments.

To understand the reach and scale of FDI in New Hampshire, we completed an analysis of foreign firms' presence across New Hampshire industries and by employment scale. Industry, activity, and employment data for these firms were collected from publicly available sources. The North American Industry Classification System (NAICS) was used in the analysis, with accompanying industry codes identifying sectors and sub-sectors. The U.S. Bureau of Labor Statistics' firm size classes were utilized to analyze employment and scale.

The study provides insights on how international enterprise touches on various aspects of New Hampshire's business and its value chain ecosystems. Findings related to scale of operations emphasize how overseas companies are represented in the state and the ways in which international business contributes to employment in New Hampshire.

5.2.1. Analysis of Foreign Firms' Integration into a Subnational Location's Business Systems

Foreign subsidiaries are engaged in the full range of value-adding activities needed to create products and services in the state. Foreign firms' statewide representation in supply chain activities shows diversity and balanced distribution in terms of presence in manufacturing, wholesale and retail (including logistics and warehousing), and support services (such as IT and administrative services). These three key areas of value chain activities in our state are steadily covered by foreign firms' endeavors: about 20 percent of firms operate in manufacturing, 30 percent of firms in trade and logistics, and 40 percent provide support services.

Figure 6 shows foreign firms' representation across sectors and identifies percentages of firms pursuing each object of activity from a total of 361 firms operating across seven main sectors in New Hampshire. The proportions demonstrate that these firms operate across stages of value chains, with good representations in primary and support value chain activities. This indicates a high level of integration of foreign subsidiaries' activities within domestic industries. Thus, foreign firms play important roles in the vitality of state industries and are integral part of New Hampshire's business ecosystem.

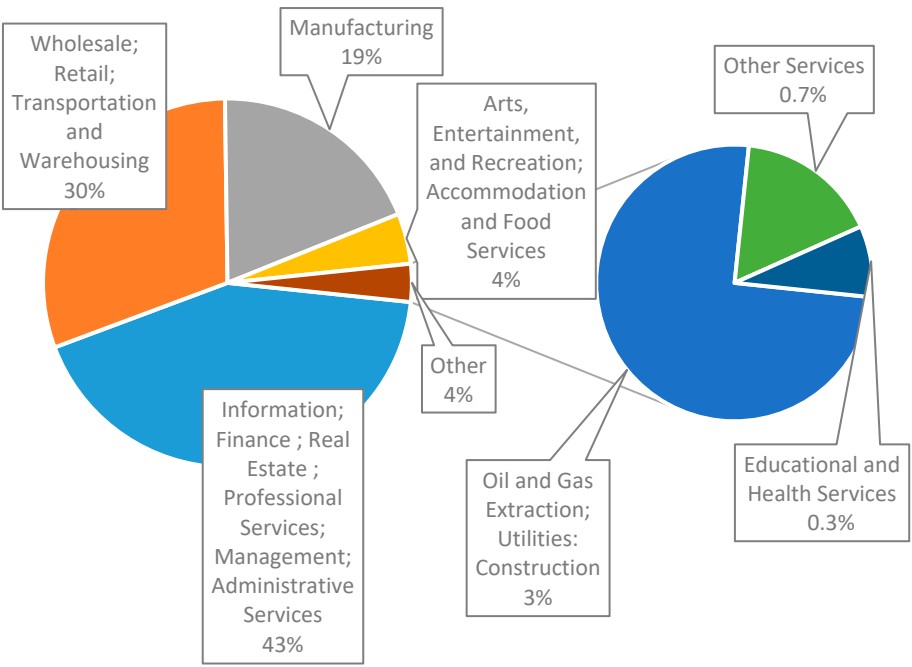

**Figure 6.** Foreign Firms' Representation Across Sectors.

5.2.2. Analysis of Scale and Employment of Foreign Operations at a Subnational Location

Most foreign firms in the state are very small. This indicates entrepreneurial scale and matches New Hampshire's overall profile as a small business state. Most foreign firms employ under 10 employees. Some large companies are operating in the state. However, as presence, they do not represent a large fraction of foreign firms. A breakdown of number of firms across size classes defined by the Bureau of Labor Statistics shows the strong manifestation of operations with smaller scale. Rather than pursuing large-scale undertakings, foreign companies may choose physical footprint and spatial reach in New Hampshire. The reduced size is likely to support connections with other local small businesses, and easy integration or collaborations with other organizations across the state.

Firm presence by employment shown in Figure 7 indicates that more than 60 percent of foreign entities in the state have under 20 staff members, and about a quarter of the total number of firms have fewer than four employees. In most cases, these small numbers are related to the fact that business operations are representative offices or branches of overseas companies. Accordingly, this information suggests that companies strive to be represented in the state even if they do not have large operations or run facilities here.

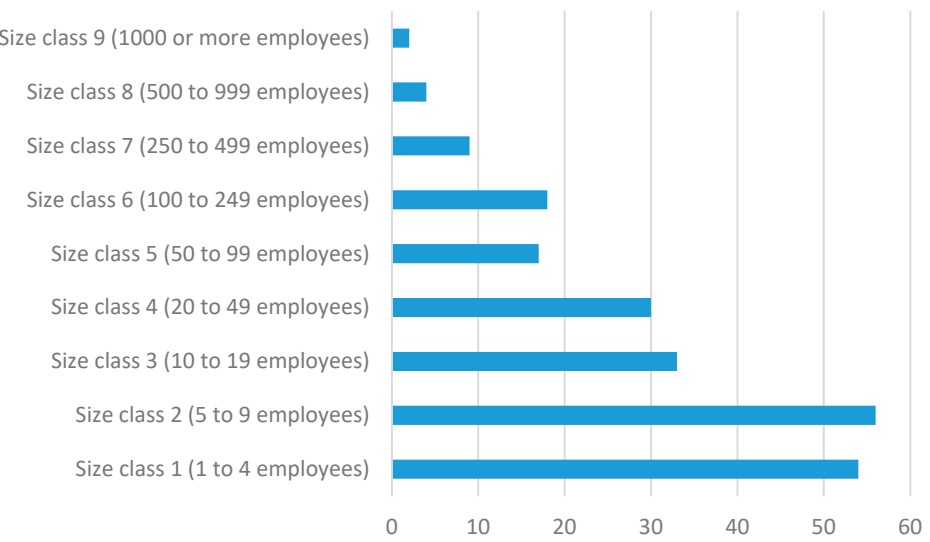

**Figure 7.** Number of Foreign Firms in Each Size Class.

The count of sectors across which foreign firms of each size class operate shows representation of foreign operations based on scale and main object of activity. These data are compiled in Table 2. There is diversity in foreign firms' sizes across New Hampshire's industry sectors. The distribution of number of employees across sectors varies. Conversely, double-digit numbers of sectors are denoted in many firm size classes.

**Table 2.** Number of Industry Sectors Represented in Each Firm Size Class.

| Firm Size Class | No. of Sectors Represented |
|---|---|
| 1 | 20 |
| 4 | 19 |
| 2 | 18 |
| 3 | 15 |
| 6 | 12 |
| 5 | 12 |
| 7 | 7 |
| 8 | 4 |
| 9 | 2 |

The comparison of foreign firms by number of employees is depicted in Figure 8 to conclude on the presence of mostly small-scale operations across the state.

Overall, foreign firms in our sample are active in six out of the nine key sectors, with no significant activities in agriculture, forestry, fishing and hunting sector and no notable representation in public administration or health services. Foreign firms' presence throughout sectors has diversity of scale, as presented in Figure 9. Markedly, foreign firms employ New Hampshire residents in a variety of industries and occupations, and support labor development across most key sectors in the state. The highest range and diversity in scale of operations can be seen in manufacturing. Similarly, the services subsectors attract foreign ventures of varying sizes. Trade and logistics facilities also vary in employment levels. Scale is more uniform in mining, oil extraction and construction subsectors. The ranges of firms' sizes may reflect a common scale of operations in particular industries, as well as strategic choices about staffing and ratios of full-time versus part-time employees. Firm size classes account only for full-time employment.

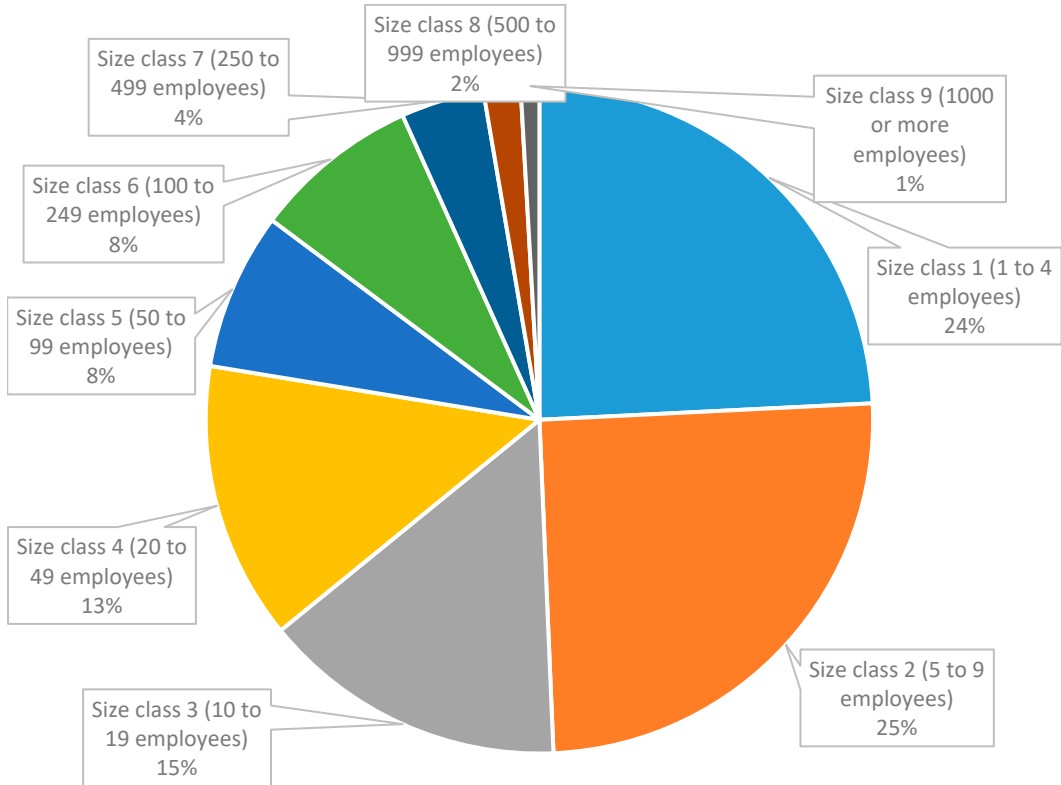

**Figure 8.** Foreign Firm Presence by Size Class.

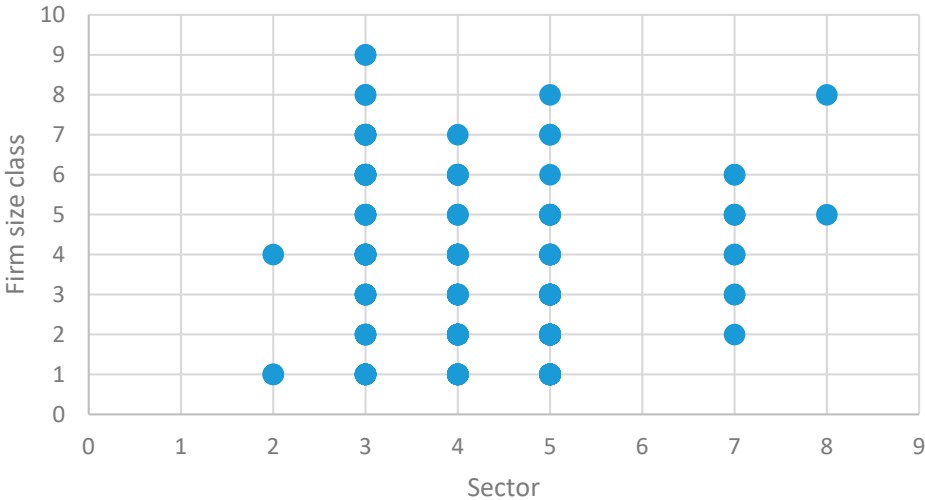

**Figure 9.** Diversity of Foreign Firms Size Across Sectors.

Note: Sector descriptions corresponding to the sector numbers on the horizontal axis are as follows:

| Sector | Description |
|---|---|
| 1 | Agriculture, Forestry, Fishing and Hunting |
| 2 | Mining, Quarrying, and Oil and Gas Extraction; Utilities: Construction |
| 3 | Manufacturing |
| 4 | Wholesale Trade; Retail Trade; Transportation and Warehousing |
| 5 | Information; Finance and Insurance; Real Estate and Rental and Leasing; Professional, Scientific, and Technical Services; Management of Companies and Enterprises; Administrative and Support and Waste Management and Remediation Services |
| 6 | Educational Services; Health Care and Social Assistance |
| 7 | Arts, Entertainment, and Recreation; Accommodation and Food Services |
| 8 | Other Services (except Public Administration) |
| 9 | Public Administration |

5.2.3. Analysis on a Subnational Location's Diversity and Concentration of Foreign Businesses across Industries and Sectors

The presence of foreign firms in New Hampshire industries shows a high level of representation in finance and insurance. Our sample identifies foreign subsidiaries with operations in at least 20 main industries recognized by the US NAICS classification. Overall, data synthesized in Table 3 indicate that foreign firms are present mainly in services, followed by strong relative numbers in trade, and, thirdly, in manufacturing.

**Table 3.** Foreign Subsidiaries Representations by Activity Types.

| Sector | Frequency |
|---|---|
| Services | 48% |
| Retail and Wholesale | 30% |
| Manufacturing | 19% |
| Utilities | 2% |
| Information | 1% |
| Construction | 1% |

In relative numbers, banking operations connected to international companies are highly noticeable in the state. This may be related to the strong networks of large multinational banks across international locations. The finding demonstrates New Hampshire individual and organization residents' access to finance and insurance systems supported by global operations. The data also indicate that retail trade has good relative representation of foreign firms, followed by manufacturing, as depicted in Table 4. The findings reveal foreign business interests in accessing local markets and expanding distribution in the state. In addition, foreign companies support manufacturing in New Hampshire, with a noticeable presence across high-tech industry segments.

**Table 4.** Distribution of Operations Across Industries in New Hampshire.

| NAICS Code | Industry | Frequency |
|---|---|---|
| 52 | Finance and Insurance | 28% |
| 44 | Retail Trade | 16% |
| 33 | Manufacturing | 13% |
| 42 | Wholesale Trade | 11% |
| 54 | Professional, Scientific, and Technical Services | 7% |
| 32 | Manufacturing | 5% |
| 56 | Administrative and Support and Waste Management and Remediation Services | 5% |

**Table 4.** *Cont.*

| NAICS Code | Industry | Frequency |
|:---:|:---:|:---:|
| 72 | Accommodation and Food Services | 4% |
| 45 | Retail Trade | 3% |
| 22 | Utilities | 2% |
| 31 | Manufacturing | 1% |
| 51 | Information | 1% |
| 48 | Transportation and Warehousing | 1% |
| 53 | Real Estate and Rental and Leasing | 1% |
| 23 | Construction | 1% |
| 81 | Other Services (except Public Administration) | 1% |
| 62 | Health Care and Social Assistance | Under 1% |
| 59 | Miscellaneous Retail | under 1% |
| 55 | Management of Companies and Enterprises | under 1% |
| 49 | Transportation and Warehousing | under 1% |
| | Total | 100% |

Frequency counts across industries classified by NAICS two-digit codes support the relative concentration in some industries, with a low number of operations in others. Although the banking sector may be more represented, manufacturing presence is relatively important. Bank affiliates throughout the state show a network of finance and insurance international connections and integration with global banking systems. Relative numbers in retail trade indicate integration in larger distribution systems and access to local customers. A good presence of manufacturing suggests that the state offers access to production labor that attracts international business interest. Other factors may also motivate manufacturing operations, such as access and development of technology, opportunities for innovation and potential for partnering with other organizations. The focus of foreign companies in and across these three key industries (finance, retail, and manufacturing) support relevancy of international activities in the state and the importance that foreign business has for the availability of financing, the growth of markets and the vitality of production in New Hampshire.

As specified in the firm's presence data, foreign-related operations in New Hampshire are highly concentrated in one (finance and insurance) industry and somewhat focused on less than a dozen industry segments, mainly in trade and services. A less notable foreign business presence spans half the industries. Industries such as utilities, some manufacturing (mechanical, physical, or chemical transformation of materials, substances, or components into new products), construction, logistics and other kinds of services do not attract major overseas presence.

Services dominate the distribution of foreign entities, with almost half of the firms operating in this sector. Across industries, services, along with retail/wholesale and manufacturing sectors are preferred by international businesses. Foreign companies are likely to participate in key projects in utilities, information technology and construction, but not with a wide-spread presence.

Figure 10 shows foreign firms' distribution in industries, with relative percentages in fields as a proportion of the count in each field in the total number of foreign firms. A high concentration in services, trade and manufacturing is evident in comparison to utilities, information, and construction.

The number of foreign-related businesses in each of the main industries depicted in Figure 11 shows variability in presence across industries, and concentration in several

fields. As mentioned above, the concentration is critical, as the top industries by presence relate directly to finance, markets, and production in the state.

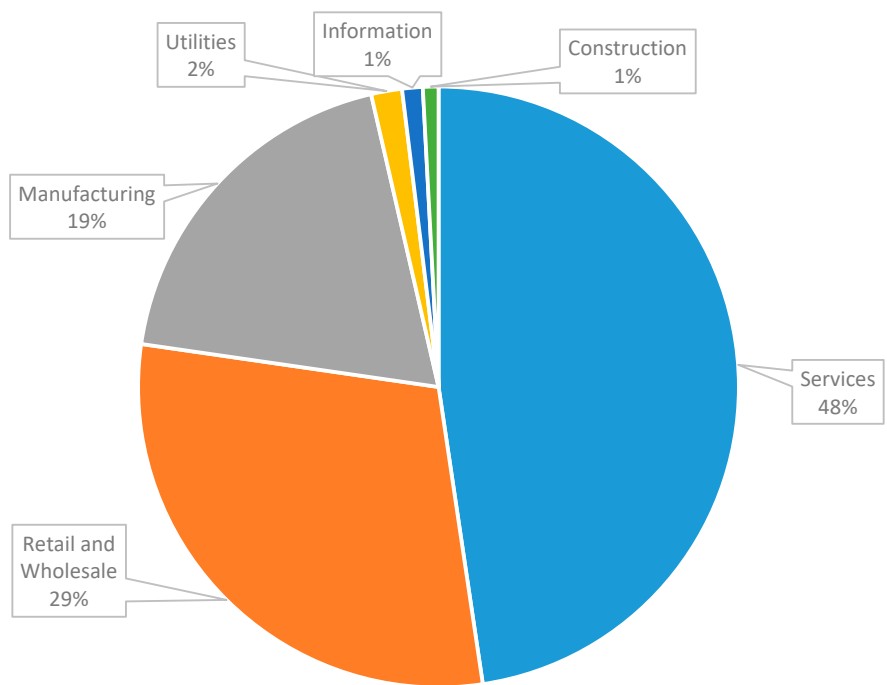

**Figure 10.** Foreign Firms' Presence by Main Object of Activity.

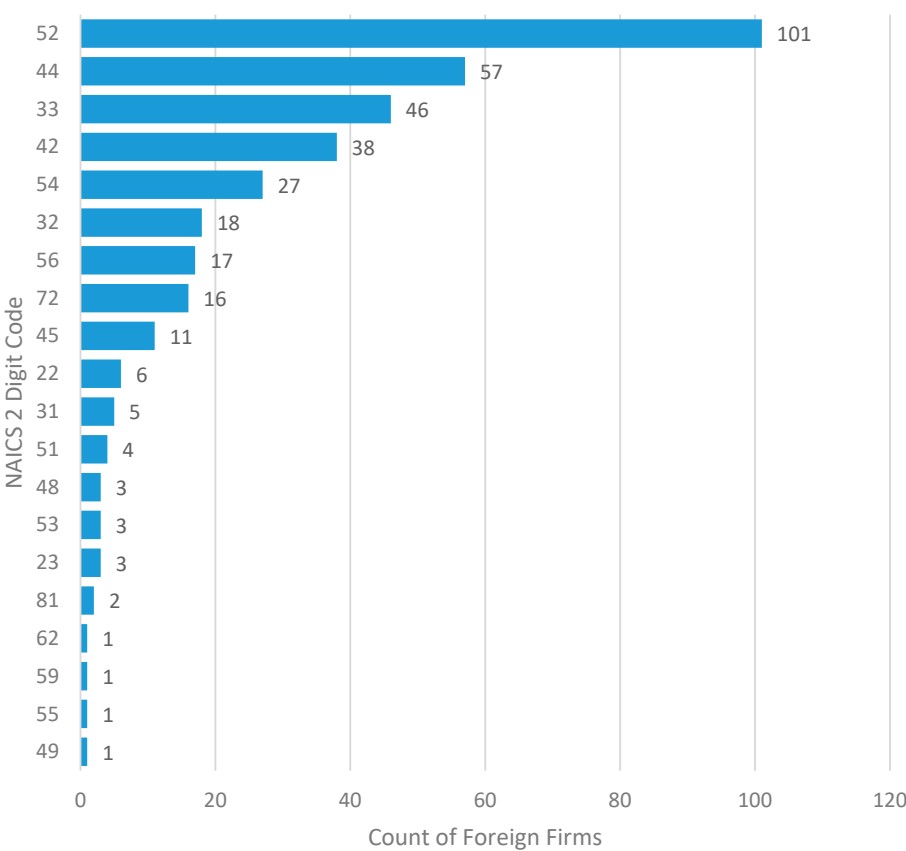

**Figure 11.** Foreign Firms' Presence by Main Industry.

### 5.2.4. Analysis on a Subnational Location's Foreign Business' Variety and Range of Purpose

Additional detail in industry presence within industry sub-sectors shows how the activities of foreign firms are directly intermingled across sectors. The data signal that activities are transversely crossing value and supply chain stages. In most industry sub-sectors, foreign firms represent less than 5 percent of the total number of foreign firms. Credit intermediation, as a sub-sector of the finance and insurance industry, points once again to a high level of concentration of foreign firms in this field. Aside from this two-digit percentage representation, all other representations in industry sub-sectors are low, in single digits. Figure 12 identifies the number of times each percentage of representation in an industry sub-sector occurs in the sample. The data indicate rich diversity of business activity.

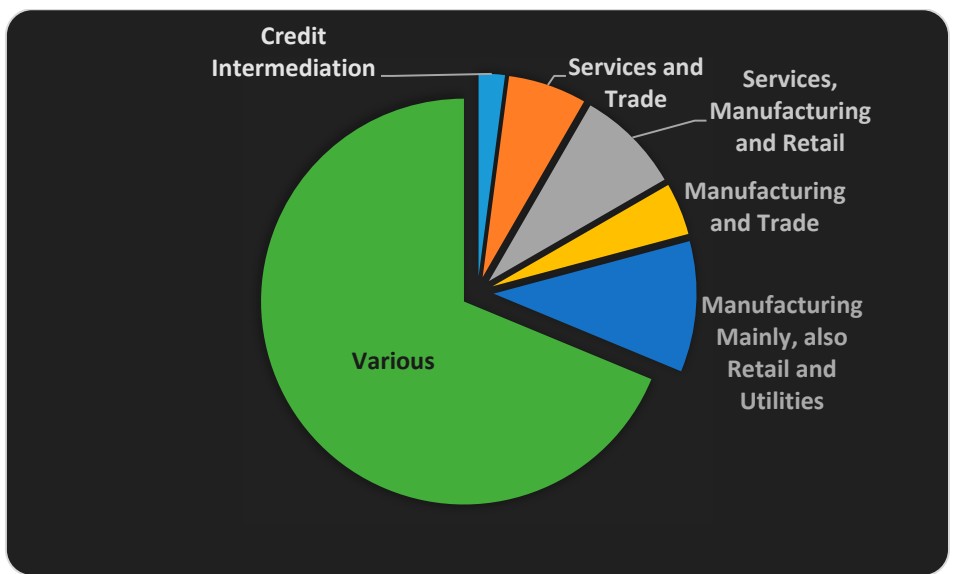

**Figure 12.** Distributions of Representation Percentages Across Industry Sub-Sectors.

The subsector distribution of firm presence shows concentration but also a more widespread range of foreign activities within each industry. The distribution signals relevant variety and range of purpose. The rich and varied presence within each industry suggests that the state offers a multitude of motivations for overseas companies. The range of industry sub-sectors may point to the vigor of international firms' business networks and relationships. Foreign firms' presence may be one compelling factor in some business leaders' orientation towards the state. A diverse international business presence may attract more attention from other overseas companies.

As shown in Table 5, the proportion of firms in each industry sub-sector indicates that there is an important presence within most industries. While some of the counts for individual sub-sectors are low, across the industry, these numbers suggest diverse activities and good coverage of the industry's many types of endeavors.

The diverse presence of foreign firms within industries once again demonstrates the high level of integration that foreign businesses have with a wide range of lucrative activities in the state. For successful operations, it can be expected that these firms collaborate and partner with local, domestic, and non-domestic companies and institutions and are an intricate part of New Hampshire business.

The data in Figure 13 confirm the foreign business presence concentration in the credit intermediation activities, followed by strong relative representations in professional, scientific, and technical services sector, in merchant wholesalers of durable goods activities, and in operation of stores. The widespread distribution in other industries shows the span and reach of foreign presence in the state's manufacturing sector across many industries sub-sectors.

**Table 5.** Foreign Presence (in Frequency) by Sub-Sector.

| NAICS Code | Industry Description | Frequency |
|:---:|:---:|:---:|
| 522 | Credit Intermediation and Related Activities | 26% |
| 541 | Professional, Scientific, and Technical Services | 7% |
| 423 | Merchant Wholesalers, Durable Goods | 7% |
| 448 | Clothing and Clothing Accessories Stores | 7% |
| 561 | Administrative and Support Services | 4% |
| 334 | Computer and Electronic Product Manufacturing | 4% |
| 721 | Accommodation | 4% |
| 445 | Food and Beverage Stores | 4% |
| 333 | Machinery Manufacturing | 3% |
| 424 | Merchant Wholesalers, Nondurable Goods | 3% |
| 454 | Nonstore Retailers | 2% |
| 326 | Plastics and Rubber Products Manufacturing | 2% |
| 332 | Fabricated Metal Product Manufacturing | 2% |
| 339 | Miscellaneous Manufacturing | 2% |
| 221 | Utilities | 2% |
| 325 | Chemical Manufacturing | 2% |
| 524 | Insurance Carriers and Related Activities | 1% |
| 518 | Data Processing, Hosting, and Related Services | 1% |
| 446 | Health and Personal Care Stores | 1% |
| 335 | Electrical Equipment, Appliance, and Component Manufacturing | 1% |
| 443 | Electronics and Appliance Stores | 1% |
| 327 | Nonmetallic Mineral Product Manufacturing | 1% |
| 444 | Building Material and Garden Equipment and Supplies Dealers | 1% |
| 523 | Securities, Commodity contracts, and Other Financial Investment | 1% |
| 447 | Gasoline Stations | 1% |
| 531 | Real Estate | 1% |
| 441 | Motor Vehicle and Parts Dealers | 1% |
| 722 | Food Services and Drinking Places | 1% |
| 312 | Beverage and Tobacco Product Manufacturing | 1% |
| 311 | Food Manufacturing | 1% |
| 485 | Transit and Ground Passenger Transportation | 1% |
| 336 | Transportation Equipment Manufacturing | 1% |
| 451 | Sporting Goods, Hobby, Book and Music Stores | 1% |
| 812 | Personal and Laundry Services | Under 1% |
| 621 | Ambulatory Health Care Services | Under 1% |
| 562 | Waste Management and Remediation Services | Under 1% |
| 236 | Construction of Buildings | Under 1% |
| 488 | Support Activities for Transportation | Under 1% |
| 238 | Specialty Trade Contractors | Under 1% |
| 313 | Textile Mills | Under 1% |
| 322 | Paper Manufacturing | Under 1% |
| 594 | Miscellaneous Shopping Goods Stores | Under 1% |
| 453 | Miscellaneous Store Retailers | Under 1% |
| 321 | Wood Product Manufacturing | Under 1% |

**Table 5.** *Cont*.

| NAICS Code | Industry Description | Frequency |
|:---:|:---:|:---:|
| 331 | Primary Metal Manufacturing | Under 1% |
| 811 | Repair and Maintenance | Under 1% |
| 237 | Heavy and Civil Engineering Construction | Under 1% |
| 492 | Couriers and Messengers | Under 1% |
| 551 | Management of Companies and Enterprises | Under 1% |
| | Total | 100% |

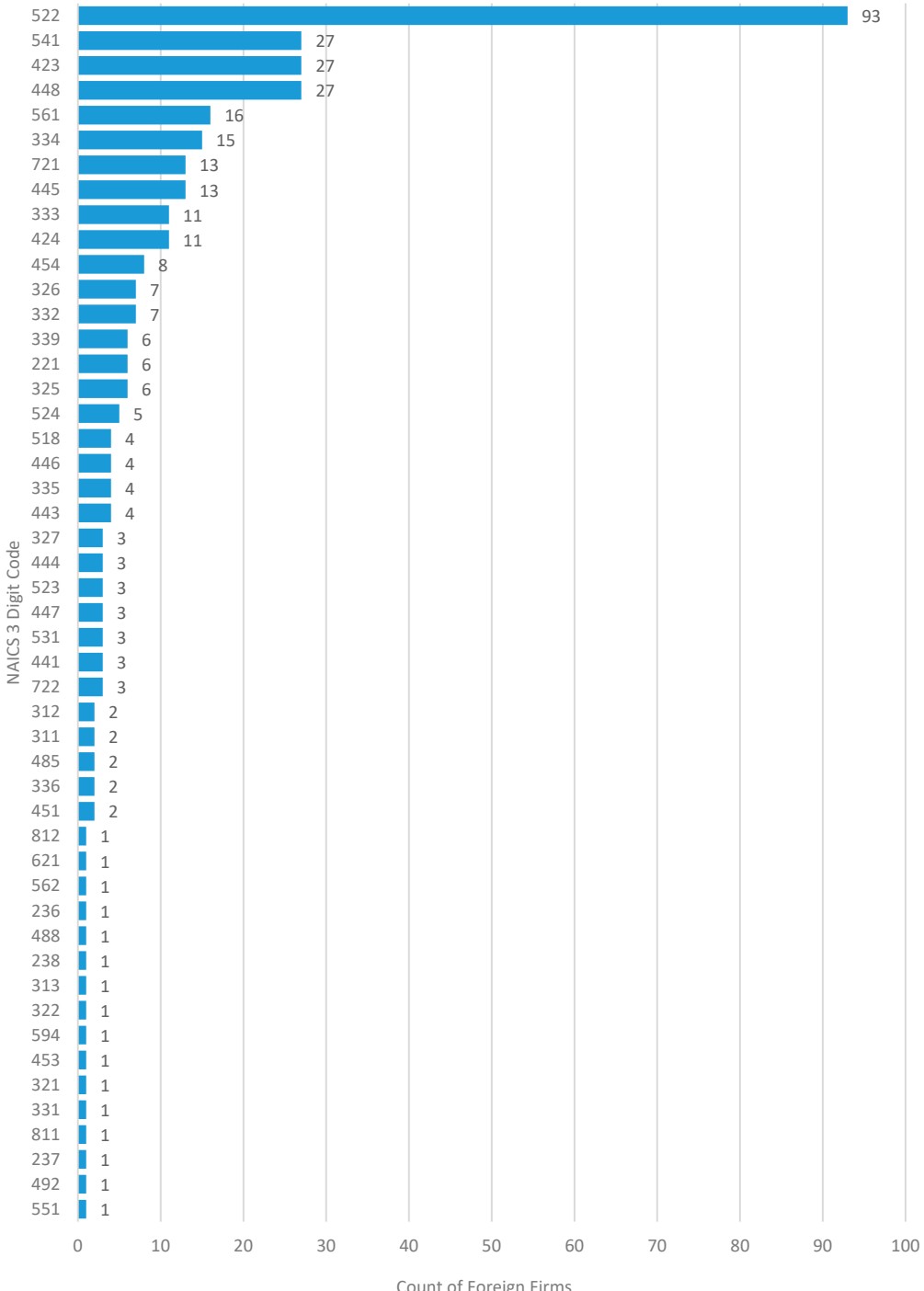

**Figure 13.** Foreign Presence (Count of Foreign Firms) by Sub-Sector.

Our exploration of industry and value chain data for the in-focus subnational location shows support for the second hypothesis anticipating that at a subnational location where foreign businesses are integrated with domestic businesses FDI-related activities are likely to remain strong post-disruption.

*5.3. FDI Presence and Resiliency*

The investigation of and support for the third hypothesis can be found in pertinent data on multinational companies' subsidiaries' presence in the state of New Hampshire. This subnational location is considered in its sources of FDI, with particular attention to FDI origins and inflows distributions across geographical localities and industries.

5.3.1. Country of Origin Analysis of FDI at a Subnational Location

In 2022, the State of New Hampshire attracted the FDI from a total of 176 parent companies headquartered in 23 countries. Figure 14 below presents a heat map for the FDI countries of origin in terms of their count of foreign subsidiaries in New Hampshire. The North America is the most heated area, followed by Europe, East Asia, Middle East, and Australia.

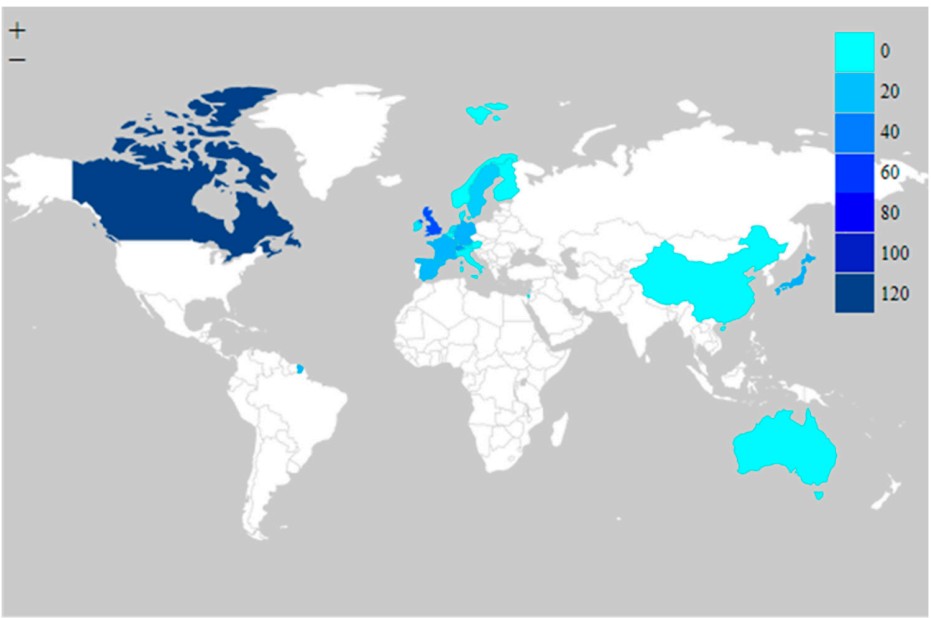

**Figure 14.** Country of Origin for FDI in New Hampshire in 2022, by Count of Subsidiaries.

Figure 15 below scrutinizes the foreign subsidiaries in New Hampshire by their countries of origin. Canada has a dominating lead with 118 (or one third) out of the 361 foreign subsidiaries. Eight other countries each have more than 10 subsidiaries in New Hampshire, among which 7 are European countries led by the United Kingdom with 53 (or 15 percent) subsidiaries and Switzerland with 30 (or 8 percent). A total of 4 countries in the Pacific Coast of Asia (Japan, China, Korea and Singapore) contribute 29 (or 8 percent) subsidiaries, primarily due to Japan, who solely has 25 (or 7 percent) subsidiaries. Israel in the Middle East and Australia are the origins for 3 and 2 of New Hampshire's foreign subsidiaries, respectively.

A further examination of the countries of origin for foreign subsidiaries in New Hampshire is conducted to reveal in what counties these parent companies invest. As illustrated in Figure 16 below, Canada is the only country of origin whose companies have foreign subsidiaries in all of the ten New Hampshire counties, among which Hillsborough and Rockingham capture one third and 30 percent of the Canadian subsidiaries, respectively. The British companies have foreign subsidiaries in eight counties of New Hampshire (except for Coös and Sullivan). The parent companies in the Netherlands (Switzerland, Bermuda)

have subsidiaries in seven (six, five) counties. Germany, France, Sweden, and Denmark each owns subsidiaries in four New Hampshire counties. All of these countries of origin choose either Hillsborough or Rockingham as their preferred investment destination.

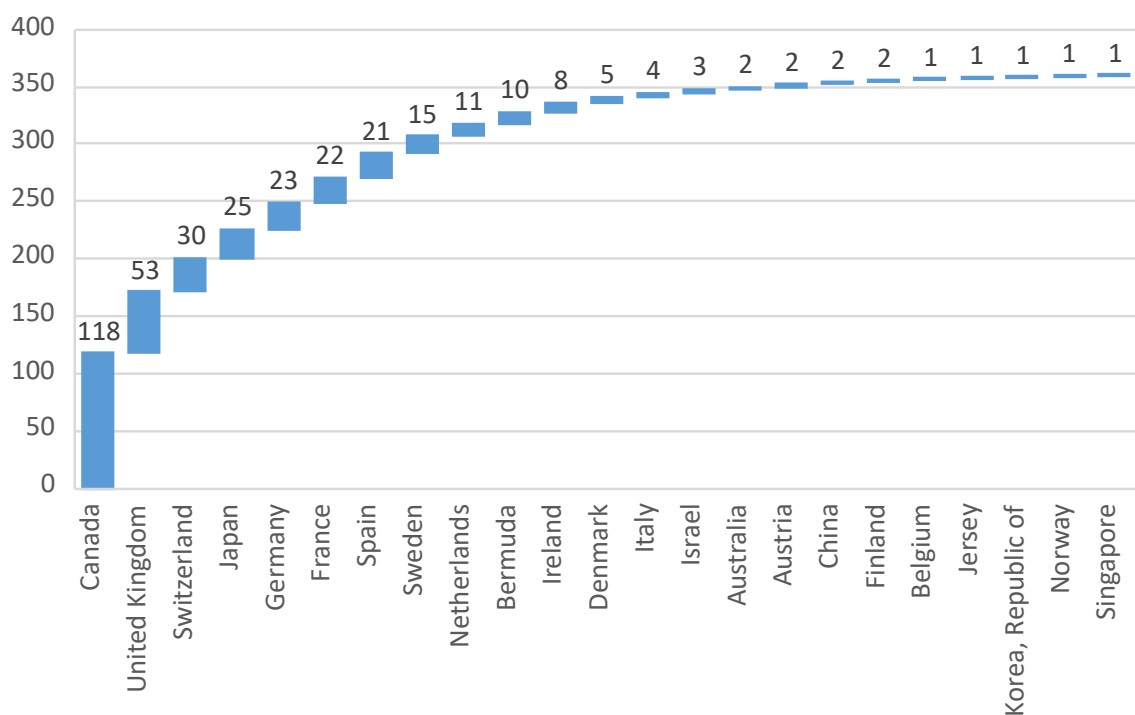

**Figure 15.** Count of Foreign Subsidiaries in New Hampshire in 2022 by Country of Origin.

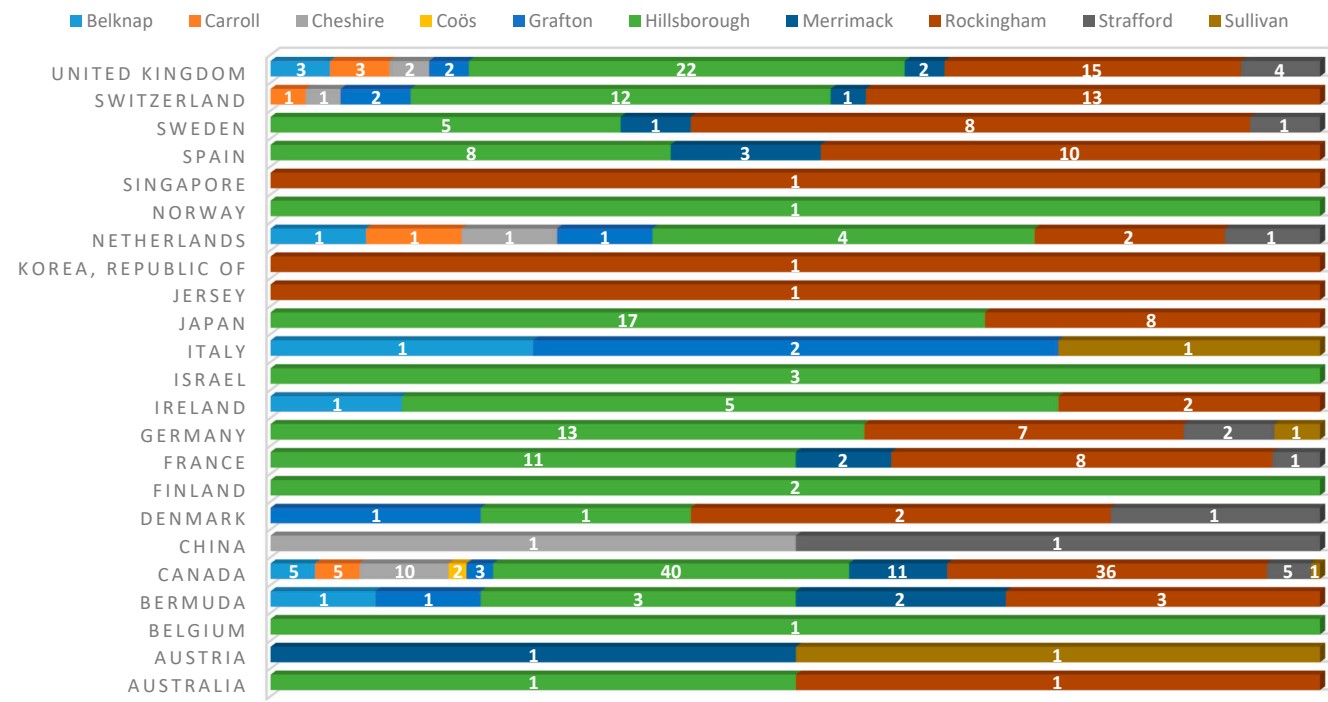

**Figure 16.** Countries of Origin for Foreign Subsidiaries in New Hampshire Counties.

5.3.2. Analysis of Intersubnational Level Foreign Businesses Presence

When the foreign subsidiaries are scatter plotted among the counties in New Hampshire, a pattern of FDI agglomeration is revealed. The bubbles in Figure 17 vary in size

because they measure the count of foreign subsidiaries for each county. The foreign subsidiaries cluster along the Massachusetts–New Hampshire border and the coast area. They spill over to the central part of the state. The FDI presence in the North Country is attributed to the border effect from Canada.

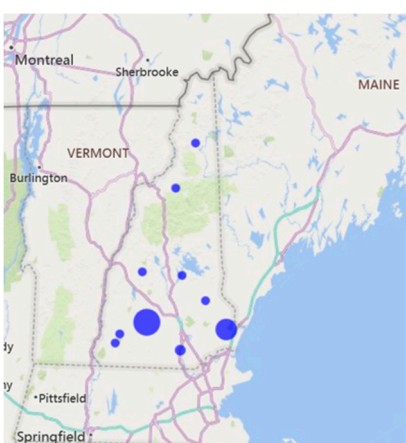

**Figure 17.** Distribution of Foreign Subsidiaries in New Hampshire Counties in 2022.

The foreign subsidiaries are unevenly distributed among all of the ten New Hampshire counties, as illustrated in Figure 18 below. The two southern counties, Hillsborough and Rockingham, have a dominating share of three quarters of all foreign subsidiaries. Adding the 4 percent share of Cheshire, the state's southern border owns nearly 80 percent of all foreign subsidiaries. The four central counties, namely Merrimack, Strafford, Belknap and Carroll, benefit from the FDI spillover and capture a total of 17 percent of the state's foreign subsidiaries. Along the state's west boundary, three counties (Grafton, Coös and Sullivan) account for the rest 5 percent of the pie.

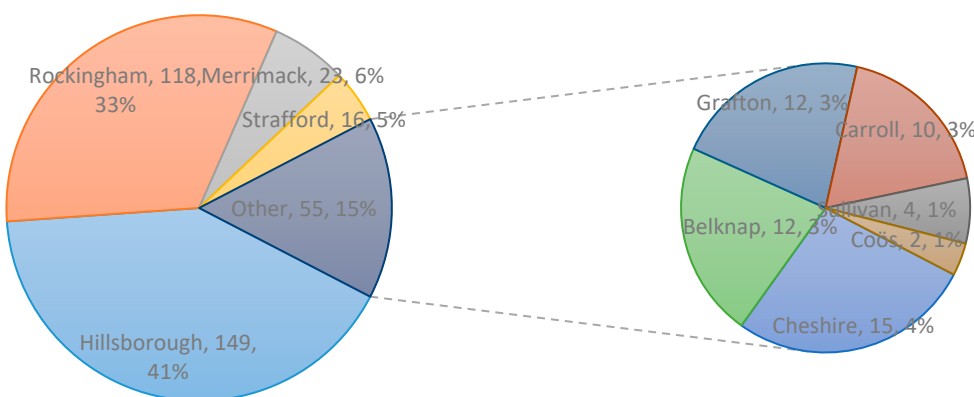

**Figure 18.** Share of FDI in New Hampshire Counties in 2022, by Count of Foreign Subsidiaries.

Figure 19 below reveals the country of origin for the foreign subsidiaries in each county. Hillsborough's foreign subsidiaries come from 17 countries. The top 5 countries of origin are Canada (27 percent), the United Kingdom (15 percent), Japan (11.4 percent), Germany (9 percent), and Switzerland (8 percent). The top 4 of Rockingham's 16 countries of origin are Canada (31 percent), the United Kingdom (13 percent), Switzerland (11 percent), and Spain (8.5 percent). Merrimack and Strafford each has 8 countries of origin for their foreign subsidiaries, and both have Canada as their number 1 source of FDI, followed by Spain (the United Kingdom), and the United Kingdom (Germany), in the top 3, respectively. Grafton, Belknap, Cheshire, and Carroll each has 7, 6, 5, and 4 countries of origin, and they all have Canada and the United Kingdom listed as the top two FDI sources. The foreign subsidiaries

in Sullivan are evenly originated from Austria, Canada, Germany, and Italy, while Canada is the sole country of origin for the foreign subsidiaries in Coös.

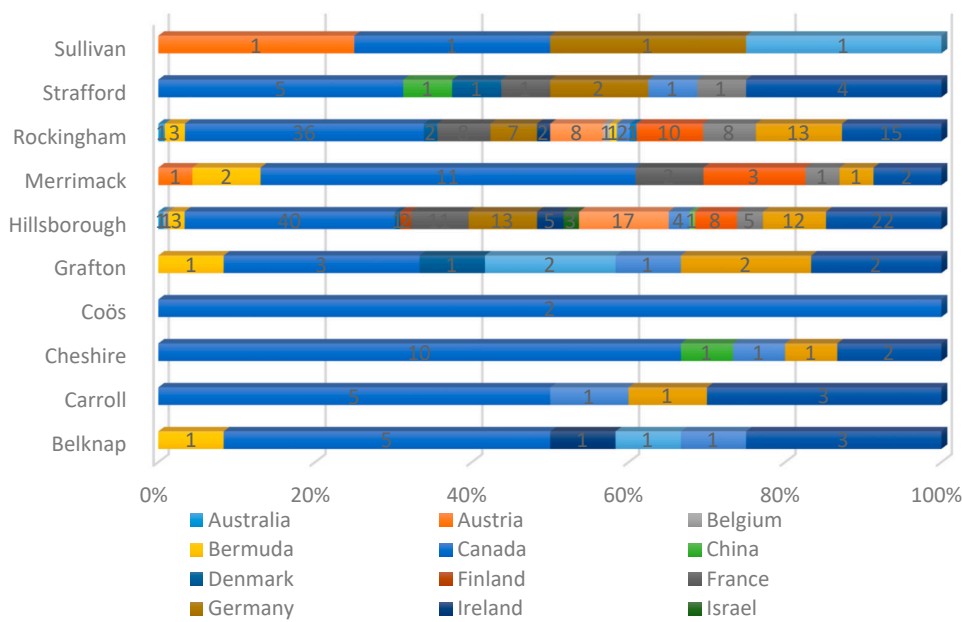

**Figure 19.** Foreign Subsidiaries in New Hampshire Counties, by Country of Origin.

When the New Hampshire counties are profiled by their foreign subsidiaries across the value chain activities, a geographic pattern in the relative importance of value chain activities is revealed. The foreign subsidiaries in the southern counties (Hillsborough, Rockingham, Merrimack, Strafford, and Cheshire) are dominated by the service providers. A main reason is their proximity to Massachusetts which is the economic center of the region. The proportion of distributors (retailers and wholesalers) among the foreign subsidiaries significantly increases and is almost the same with service providers for the inner center counties (Belknap, Grafton, and Carroll). This highlights the importance of central New Hampshire as the shopping center for the northern area and as their gateway to the bigger southern market. For the North Country and Sullivan, the share of producers is equal to the share of service providers among their foreign subsidiaries, suggesting the potential to grow the manufacturing FDI in this area. Figure 20 below details the data.

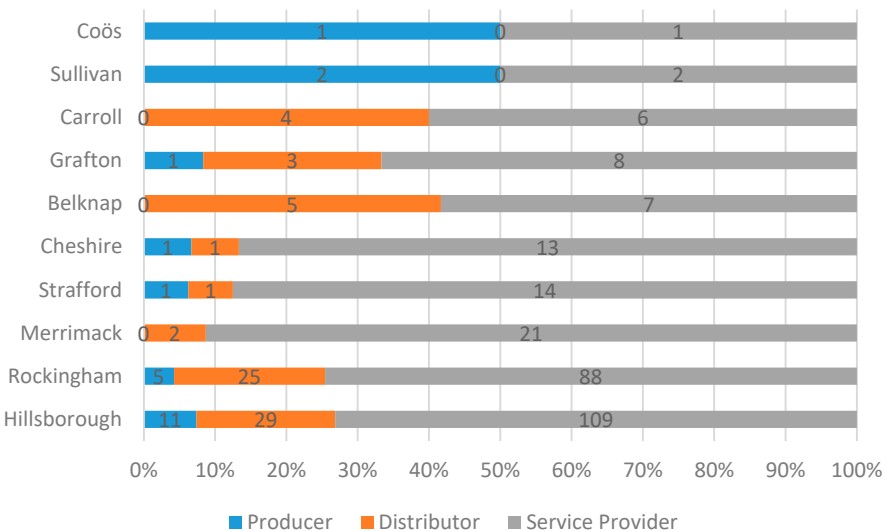

**Figure 20.** Foreign Subsidiaries in New Hampshire Counties in 2022, by Value Chain Activity.

### 5.3.3. Industry and Value Chain Activity Analysis of FDI Originators at a Subnational Location

The parent companies of New Hampshire's foreign subsidiaries engage in a total of 14 NAICS 2-Digit industries across all of the four economic sectors. As shown in Figure 21 below, near 60 percent of these parent companies are in the Secondary Sector, followed by the Tertiary Sector (25 percent) and the Quaternary Sector (17 percent). Figure 22 further illustrates the distribution of these parent companies by industries. In the Secondary Sector, the manufacturing industry (NAICS 31-33), with a 55 percent share, is the largest single industry of New Hampshire subsidiaries' parent companies, followed by the utilities industry (NAICS 22, 2 percent) and the constructions industry (NAICS 23, 0.5 percent).

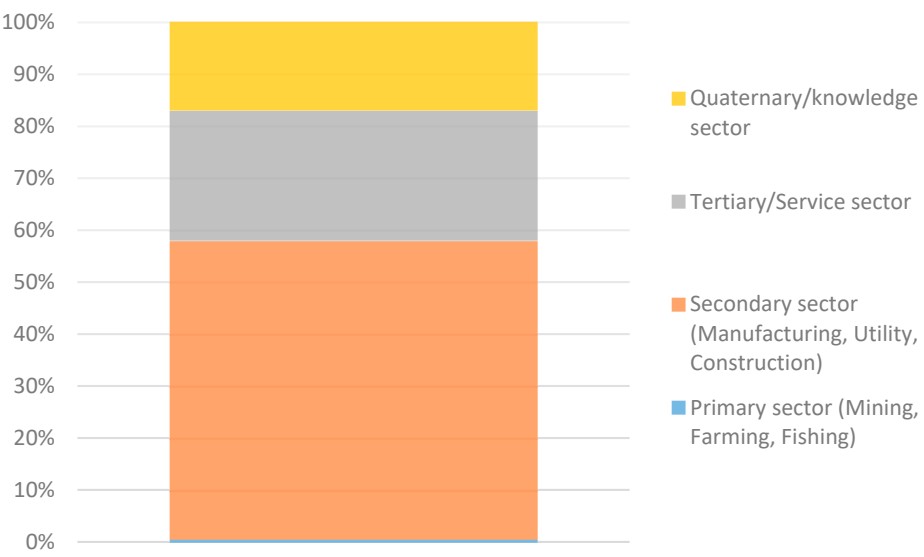

**Figure 21.** Sectors of Parent Companies of Foreign Subsidiaries in New Hampshire.

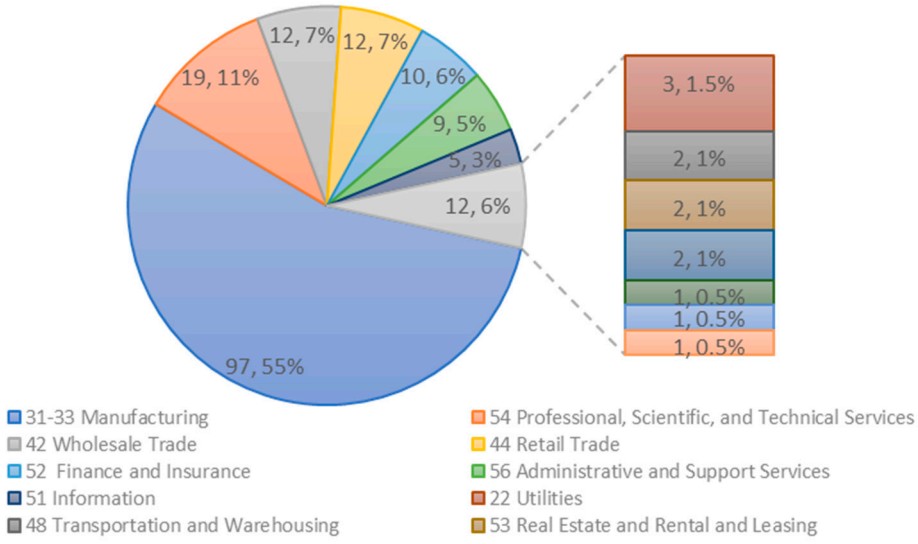

**Figure 22.** Industries of Parent Companies for FDI in New Hampshire, by NAICS 2-Digit Code.

In the Tertiary Sector, the wholesale trade industry (NACIS 42) and the retail trade industry (NACIS 44) each has 7 percent of the parent companies, followed by the finance and insurance industry (NAICS 52, 6 percent) and the information sector (NAICS 55, 3 percent). The Quaternary Sector is also referred to as the Knowledge Sector, in which the professional, scientific, and technical services industry (NAICS 54) is the second largest single

industry (next to the manufacturing) with a share of 11 percent of all parent companies, followed by the administrative and support services industry (NAICS 56, 5 percent) and the management of companies and enterprises industry (NAICS 55, 1 percent). Only one parent company from the mining, quarrying, and oil extraction industry (NAICS 21) is in the Primary Sector.

The foreign subsidiaries in New Hampshire contribute to all value chain activities of the state's economy. As shown in Figure 23 below, three quarters of them provide services to consumers and businesses, 19 percent facilitate the distribution of products via retail and/or wholesale trade, and 6 percent produce value added finished products from the raw materials.

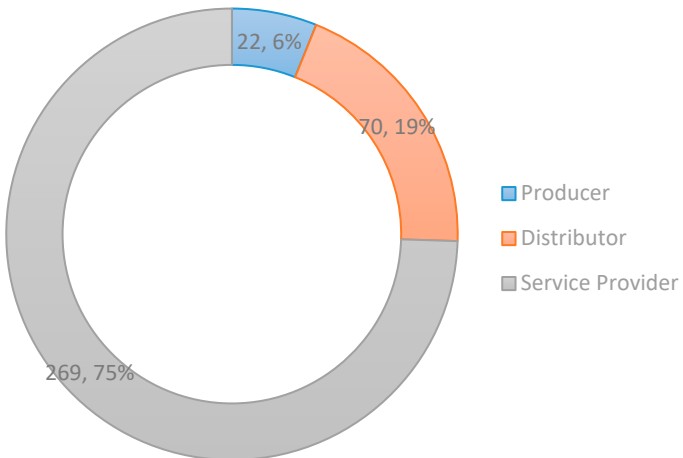

**Figure 23.** Value Chain Activity of Foreign Subsidiaries in New Hampshire in 2022.

Figure 24 below describes the value chain activities of New Hampshire foreign subsidiaries by their county locations and reveals an FDI agglomeration pattern. The two southern border counties (Hillsborough and Rockingham) have a dominant lead with a joint share of around 75% in all of the three categories. From this FDI cluster, 18 percent of the service providers spill over to the neighboring counties (Merrimack, Strafford and Cheshire), 17 percent of the distributors further spill over to the inner center counties (Belknap, Grafton, and Carroll), and 18 percent of the producers even spill over to the state's northwest border (Sullivan, Coös, and Grafton).

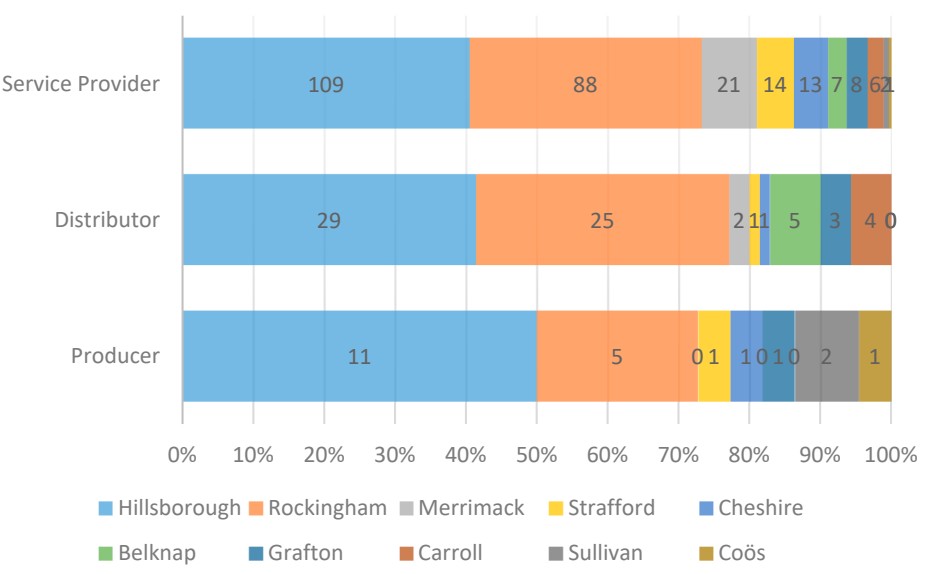

**Figure 24.** Value Chain Activity of Foreign Subsidiaries in New Hampshire in 2022, by Counties.

Our data on FDI presence across the in-focus subnational location and the origins of FDI show that, at locations that have deep connections with the FDI origins and where there are already agglomerations of foreign businesses, FDI presence is more resilient to disruption. This supports our third hypothesis.

### 5.3.4. Impact of COVID-19 Pandemic Disruption: Comparison between 2018 and 2022 at a Subnational Location

Wu and Wright (2018) provided an overview as well as a deep dive into data on foreign business presence in the state of New Hampshire. At that time, the authors were remarking on the large number of industries where foreign firms had a presence, the concentration of international activity in finance, and the fact that most business entities with international ties were small in scale.

Four years on and at the tail end of a pandemic that brought significant global disruption, the data show the resilience of foreign businesses in New Hampshire. These firms continue to be widely represented across industries. In fact, the number of industry sub-sectors across which foreign firms in New Hampshire operate is the same as in the past half-decade. Finance and insurance still represent the industry with the most relative international representation. The percentage of foreign firms that are in finance and insurance is almost the same as in pre-pandemic times. Just like in pre-2020s years, overseas companies in the state are represented by small firms with fewer than 20 employees. The pre-pandemic study found that firms with international connections are represented at many stages of the supply chain and across many industries. The proportions of service providers, producers and distributors are comparable with the pre-pandemic distributions. This indicates the closeness of the integration of foreign firms into local supply and value chains, giving these firms a basic role in the development of the local ecosystem. The data also support the idea that foreign firms continuously add value to products and services in the state. The distribution of FDI presence and activities within this subnational location are very similar to the pre-disruption dispersal. The parent company's characteristics and objects of activity match those of the pre-pandemic years.

At our in-focus subnational location, we find resiliency of FDI and consistent international businesses' interest post-disruption. There is diverse representation across industry sectors and sub-sectors. International business permeates most parts of the state's business and economic development. Second, overseas links exist along and across value and supply chains. The data illustrate how international business and global connections persist and add value to products and services. Foreign firms partner with domestic entities and support the viability of activities that produce and distribute value. Finally, international business is represented by firms that are comparable in size to local companies. Foreign firms in New Hampshire are just like any other firms in the state: diverse in object of activity, integrated in local and global supply chains, and entrepreneurial. The "foreignness" associated with these firms should not evoke a sense of difference or separation from local business development. Just like all other firms, international firms continue to provide employment in many industries and across areas of expertise, strive for innovation and resilience in supply chains, and add to the entrepreneurial business mix at the subnational location (Umiński and Borowicz 2021).

## 6. Conclusions

In the present study, we advocate for a new and unique perspective to understanding FDI recovery and resiliency. We juxtapose FDI, the presence of companies generating FDI and related activities at the subnational location to national levels and patterns. While the national data offer an integrative view of FDI, we show a new view on FDI amounts and behaviors at subnational location. Our paper proposes a new perspective on disruption and means to anticipate the effects and outcomes of disruption. The resulting insights can help economic development professionals and foreign and domestic local businesses to ready

their organizations for promoting the subnational locality to other overseas companies and for finding relevant partners now that normalcy is returning.

The present study can help economic development experts and business leaders at a subnational location in three critical areas: to develop an understanding of the areas in which foreign businesses contribute to the state's domestic product, to gain insights on the potential of finding partners with overseas affiliations in particular sectors and sub-sectors in the state, and to restore promotion and development plans to pre-pandemic levels via partnerships with local foreign businesses.

Beyond the evidence supporting our hypotheses that aim to clarify the effects and outcomes of the disruption on FDI and FDI-related activity at a subnational location, we also find meaningful themes in our in-focus local data on FDI. These findings present important facets of FDI presence and the COVID-19 pandemic effects on this presence. Our data show that FDI-generating firms may be matching the profile of domestic businesses at a location, such as entrepreneurialism. These firms may actively look to integrate in the business ecosystem. Most multinational firms are at a subnational location for the long run. Despite the disruption caused by the pandemic, the current study finds that FDI and multinational representations across activities and sectors can remained the same as before the pandemic. Subnational FDI flows to a diversity of activities and across sectors, even if there may be key investors or projects that make up a lot of investments. As presence, FDI at a subnational level can be expressed in extensive operations, but also through representative offices and small branches charged with distribution of international or global products and brands.

Common understanding has been that the pandemic drove many leaders to focus on solving immediate interruptions and on dealing with the challenges of day-to-day operations when supplies and staff are limited. However, the data examined in this study may signal that, in the face of disruption, foreign businesses have continued to maintain good presence at subnational locations. Now may also be the time to renew commitments to building a culture of integration in the local business environment. Overall performance and supply-chain management improve when parties have deeper, non-transactional, relations. The potential for optimization of relationships after the pandemic is high and more decision makers should place their efforts towards it.

The analyses provided in this study strive to serve leaders with research, advice, and inspiration for performance-targeted solutions in both economic development and business expansion. The work combines the disciplines of research, economics, and strategy to empower people who can make local economic and business growth possible.

## 7. Limitations and Extensions

In our study, we were able to verify three hypotheses that identify how FDI may recover after disruption and the themes that appear to identify resilient FDI. We used data from one subnational economy to provide context and support for our hypotheses. The local data descriptions and comparisons to national data offer insights on FDI from many perspectives, such as new and existing, presence, related activities, and origination. The methodology allows for pertinent FDI depiction and meaningful discussion on its upturn and strength. Nevertheless, this type of investigation cannot test hypotheses or connect statistically factors related to the key issues considered. Our data are also limited to one subnational location, which may restrict details of all possible patterns and themes. Our study's findings cannot be generalized, but rather provide a step forward and a robust framework for future analyses.

Whilst our research is unique in linking the disruption caused by the pandemic and the resilience and characteristics of FDI presence at a subnational level, the recency of the shocks and the ongoing recovery of business mean that many avenues for research remain open in the future. Studies exploring the hypotheses at various subnational locations would add to an understanding of what leads to FDI resiliency and how to support it with economic policy and good business decisions (Hutzschenreuter et al. 2020). Modelling the

relationship between local institutions and business fundamentals and existing features of FDI on the one hand and FDI resilience on the other hand could uncover the strength of the association between characteristics of a subnational location and FDI strength. A large dataset covering such variables across subnational locations could test the relevant hypotheses. Comparing data among varying localities could be another way to extend the investigation. Such comparisons could reveal intra-national patterns of FDI strength and map out where and how FDI and FDI related activities recover after disruption. It would be worthwhile to understand more about FDI and its resilience in the context of the subnational level. The COVID-19 pandemic has offered a window into what the consequences of disruption may be on international business. As the world continues to experience shocks and uncertainty, a deep understanding of all facets of resilience and how to create it is important for both business decision-making and regional development policy (Nguyen and Lee 2021).

**Author Contributions:** Conceptualization, R.W. and C.W.; methodology, R.W. and C.W.; software, R.W. and C.W.; validation, R.W. and C.W.; formal analysis, R.W. and C.W.; investigation, R.W. and C.W.; resources, R.W. and C.W.; data curation, R.W. and C.W.; writing—original draft preparation, R.W. and C.W.; writing—review and editing, R.W. and C.W.; visualization, R.W. and C.W.; supervision, R.W. and C.W.; project administration, R.W. and C.W. All authors have read and agreed to the published version of the manuscript.

**Funding:** This research received no external funding.

**Data Availability Statement:** The raw data supporting the conclusions of this article will be made available by the authors on request.

**Conflicts of Interest:** The authors declare no conflict of interest.

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
