# Peer review of "Is Foreign Direct Investment Resilient Post the COVID-19 Pandemic? The Case of a Subnational Economy"

_jrfm, doi:10.3390/jrfm17010021_

Round 1
Reviewer 1 Report
Comments and Suggestions for Authors
The topics discussed in the article are current. To date, the effects of the pandemic are felt in many aspects - both economic and social. Therefore, the problem raised seems to have a strong justification.
The article uses current and latest publications. The number of publications 27 seems to be small. However, it is properly selected. Therefore, this number of publications can be accepted.
The results presented are detailed. They were presented clearly.
The structure of the article is unquestionable.
The study of the impact of the COVID-19 pandemic on FDI was based on the state of New Hampshire. The results are interesting and valuable. However, the conclusions should indicate the limitations of the study. In addition, information related to the prospects of the study should be added. Briefly add whether research can be continued in the future. If so, what should this sequel look like? Especially since the article states that the proposed more detailed than national approach may help economic development experts and business leaders. Therefore, it is worth elaborating on this topic.
Additionally, it is worth considering whether it would be worth adding a discussion in the conclusions section.
The article can be published after editing the conclusions section.
Reviewer 2 Report
Comments and Suggestions for Authors
1-Please add values to the vertical axis (in Figure 1) and please place the source under the figures.
2- Please ensure that all graphs maintain a consistent format. Additionally, kindly reorganize all tables, as their current structure is not optimal.
3-The limitations of the study and recommendations for future studies should be explained in the conclusion section.
Reviewer 3 Report
Comments and Suggestions for Authors
Dear Authors,
The content you presented is interesting and has cognitive value. The issue of foreign direct investment, its local specificity, the consequences of Covid for FDI or FDI resilience is important, current and worth analyzing. In this sense, I evaluate the text positively. However, the text structure needs to be improved. Now the presented content is a report and not a scientific article (by the way, this is also the name of the text in line 644).
The text lacks a clearly defined purpose of "report-article" and a description of research methods. Reading the text, I can guess that the scientific purpose of the "report-article" is to indicate the importance of FDI in local development and to determine how resistant FDI at the local level is to economic disruptions (covid-19). The purpose in an empirical sense is to present the specifics of FDI in New Hampshire, US. However, these are my guesses because it is not mentioned in the text. This must be corrected.
The research methods used are a simple presentation of data, based on which only directions for further research can be indicated, not hard conclusions. The research methodology was completely omitted, I know that the analyzes were based on graphical analysis, but the conclusions go beyond the analysis of graphical data.
My suggestion: after re-analyzing the text, the authors should consider what the actual purpose of the analyzes was and how they planned to achieve this purpose, and then arrange the text again (in a new way). The text should lead to a final discussion and conclusions, which should include conclusions about the written text.
When you clearly specify the purpose of the article and describe the research method and procedure, I will be able to assess whether your (the authors') intentions have been achieved.
I encourage you to improve the text - the topic is really interesting.
Kind regards,
Reviewer 4 Report
Comments and Suggestions for Authors
The specific paper is an interesting presentation of FDI in a specific State of US.
However, it lacks the scientific point of view to be suitable for publication in the specific journal.
Some conclusions underlined in the paper are not strictly scientific but mainly assumptions of the authors. For instance, lines 197-201: Just with the data of only one State, we cannot derrive such general conclusions.
"It is worth noting that the U.S. and New Hampshire shared the same percentage growth in the new FDI expenditures from 2020 to 2021 but the U.S. restored and outperformed the average of its values between 2014 and 2019, while New Hampshire did not. This comparison reveals the COVID-19 pandemic had a more severe impact on some states (including New Hampshire) than the others.
The same holds in lines 315-319. There may be other reasons as well, like taxing, production cost, outscorcing etc.
- The research purpose of the paper is not clear. What is the research question? The comparison between US and New Hampshire? The New Hampshire itself? something else? The authors should specify that. The whole paper is more a (very good indeed) quantitative report concernig the presence of subsidiary companies in New Hampshire and less a research manuscript that answer specific research questions. A lot of different qualitative data exist relative to foreign firms in New Hamphsire, however are not strictly (and scienticaly related to some research specific questions)
- The resilience of FDI is not clearly demonstrated. It is logical, after a period of closed shops, manufacturing and shops, to have an increase a year after that. An the resilience is also logical, as the world returns to the past conditions (pre COVID-19). I cannot find a clear research point on that.
Reviewer 5 Report
Comments and Suggestions for Authors
1.The title is very general and does not reflect the content. Rewrite the title to be more catchy and more appealing to wider journal audience
2. The objective of the paper presented need elaborated clarifications. It is necessary to convey the motivation of the study, especially for the chosen case, hypothesis or research question well stated.
3. The introduction requires rewriting because it does not reflect the importance of the work, the technique that will be used. Figure 1 should go in the statistical section, it is a distraction since the importance of the article is not adequately understood
4. There is no serious discussion of the data and its origin. It also does not describe the technique that will be used
5. Greater depth is required in the theoretical section since it is very simple
6. on the part of the Proposition 1. The global disruption will have more severe impacts on new incoming FDI for some subnational locations than others. So, the new FDI activities in some locations will take a longer-than-national average time to restore their pre-disruption levels. I believe that not only new investments should be analyzed but also the reinvestments that many foreign companies make in the same town or region, which are sometimes more important than new investments. 7. In the second proposition, it is not discussed in the introduction and had not been mentioned, but it is also not clear how companies not only did not influence the employment they generate directly but indirectly through the productive chains that they have at the same time. inside each region. Although that does not develop it adequately. I consider that this could be the potential of the document. But it does not explore theoretically nor does it demonstrate empirically. 8.It would be convenient to show the importance of FDI in the employment level in the region since it does not raise the question of whether they are the ones that determine employment in the greatest proportion or whether it is local companies and it would be good to know not only the impact of FDI but rather how local companies, through their reinvestments, are the ones that impact more or less on the level of employment and productive chains.Author Response
Please see the attachment.

Round 2
Reviewer 1 Report
Comments and Suggestions for Authors
Corrections have been made to the article in line with the reviewers' suggestions.
The structure of the article is appropriate.
The content of each section is appropriate. References to other items from the literature are current and appropriate. The presented research results were correctly prepared. The conclusions contained in the article are properly formulated.
The article may be published.
Author Response
Thank you for your thoroughness and thoughtfulness in the review process. Your constructive feedback were all well taken for revisions and they greatly improved the quality of our study.
Reviewer 3 Report
Comments and Suggestions for Authors
Dear Authors,
the changes you make are important. The changes improved the quality of the article. It was a good idea to introduce a methodological part.
However, I do have a few suggestions to consider before publishing.
First, you presented Part "4. Propositions" and there you discussed the propositions of the study. Scientific articles do not accept propositions. Hypotheses are verified during research. This part of the article should be adapted to the scientific level of the article.
I still think it's more of a report than a scientific study, but sometimes understanding a phenomenon is important enough to be worth publishing. The topic of FDI at the local level has great cognitive value.
The second important suggestion concerns the graphics of the article. Match the figures and tables to the journal. Each figure should have a title (remove title from area figure, add title above the figure).
I hope you will make a correction. Kind regards,
Reviewer 4 Report
Comments and Suggestions for Authors
the paper is ok in its current form
Author Response

(The authors gave the same response as above.)

Reviewer 5 Report
Comments and Suggestions for Authors
no suggestions
Author Response

(The authors gave the same response as above.)

Round 3
Reviewer 3 Report
Comments and Suggestions for Authors
Dear Authors,
thank you for the corrections you made. The quality of the article has been improved. I still believe that the article only used simple methods of presenting data and that is where its value lies. Therefore, I rate the value of the article as "average". Knowledge about FDI at the local level is important, therefore, despite its methodological simplicity, the article should be published.
Pay attention to numbers and editorial requirements.
I wish you further publishing success.
Kind regards,